# Framing the Family: A Qualitative Exploration of Factors That Shape Family-Level Experience of Pediatric Genomic Sequencing

**DOI:** 10.3390/children10050774

**Published:** 2023-04-25

**Authors:** Hadley Stevens Smith, Emily S. Bonkowski, Madison R. Hickingbotham, Raymond Belanger Deloge, Stacey Pereira

**Affiliations:** 1PRecisiOn Medicine Translational Research (PROMoTeR) Center, Department of Population Medicine, Harvard Medical School and Harvard Pilgrim Health Care Institute, Boston, MA 02215, USA; 2Institute for Public Health Genetics, University of Washington School of Public Health, Seattle, WA 98195, USA; 3Center for Pediatric Neurological Disease Research, St. Jude Children’s Research Hospital, Memphis, TN 38105, USA; 4Department of Molecular and Human Genetics, Baylor College of Medicine, Houston, TX 77030, USA; 5Center for Medical Ethics and Health Policy, Baylor College of Medicine, Houston, TX 77030, USA

**Keywords:** genomic sequencing, family, utility, qualitative, psychosocial

## Abstract

Families of children with rare and undiagnosed conditions face many psychosocial and logistical challenges that may affect their approach to decisions about their child’s care and their family’s well-being. As genomic sequencing (GS) is increasingly incorporated into pediatric diagnostic workups, assessing the family-level characteristics that shape the experience of pediatric GS is crucial to understanding how families approach decision-making about the test and how they incorporate the results into their family life. We conducted semi-structured interviews with parents and other primary caregivers of pediatric patients who were evaluated for a suspected genetic condition and who were recommended to have GS (*n* = 20) or who had recently completed GS (*n* = 21). We analyzed qualitative data using multiple rounds of thematic analysis. We organized our thematic findings into three domains of factors that influence the family-level experience of GS: (1) family structure and dynamics; (2) parental identity, relationships, and philosophies; and (3) social and cultural differences. Participants conceptualized their child’s family in various ways, ranging from nuclear biological family to support networks made up of friends and communities. Our findings can inform the design and interpretation of preference research to advance family-level value assessment of GS as well as genetic counseling for families.

## 1. Introduction

Rare, genetic, and undiagnosed conditions can have substantial impacts on both the physical health and emotional well-being of the child and family. Families of children with undiagnosed conditions face uncertainty and psychosocial and logistical complexity related to their child’s health and clinical care trajectory [1,2,3]. While families’ experiences can vary widely, challenges with navigating the health care system [4,5,6,7]. and unmet informational and psychosocial needs [8]. are common across countries.

Diagnostic workups for pediatric patients increasingly include genomic sequencing (GS) [9,10,11,12]. Qualitative research exploring motivations for testing has demonstrated that parents value obtaining additional knowledge about their child’s health condition, altruistic feelings of contributing to medical science, and enhancing their own reproductive autonomy as well as that of other family members [13,14]. Additionally, some parents feel a parental duty or obligation to ensure their child is receiving the most comprehensive care [1,15]. Parents have described feelings of both hope and worry about what GS may find [16,17]. The testing process sometimes leads to feelings of overwhelm or frustration, [5]. and families have different preferences regarding the type and amount of information they want to receive [18]. GS uptake may also be influenced by logistical barriers and test-related factors such as accuracy, cost, and data privacy [16,19,20]. Factors related to parents’ own perspectives, such as their perceptions of the benefits and risks of testing and their personal attitudes toward testing [21], as well immigration status, primary language spoken [22], and level of acculturation [16], may influence their decision making about GS.

The decision to pursue GS and how families experience the process of pediatric GS may be shaped by their experiences and perspectives, including personal, sociocultural, and family factors [16,21,23]. Given the potential relevance of pediatric GS results to the patient’s family members and the role of the family in decision-making for pediatric care [20,24], it is crucial to understand the family-level characteristics that shape the experience of pediatric GS. One prevalent definition of family is relation by birth, marriage, or adoption [25], yet understanding of family often extends beyond those parameters. Conceptualizations of what a family is and who is included may differ across family structures and across biomedical, social, and cultural contexts [26,27,28]. Parents and other primary caregivers may consider various other relationships, including neighbors and friends, as a part of the child’s family [29]. Additionally, traditional roles within a family may be occupied by other relatives, such as similar-age aunts and cousins filling the role of sibling to a child with a chronic illness [27]. Each family’s unique relationships may play a role in shaping their experience of the GS process through influencing decisions on both whether to pursue testing and with whom to share results. While the parental perspectives on many aspects of the GS process have been well-studied [30], little research has focused on how family-level characteristics influence preferences for and experience of clinically indicated pediatric GS. The intersectionality of factors that shape the family-level experience of GS for a child with an undiagnosed condition is also not yet well understood.

We explore whether and how features of families affect their preferences and experience of clinically indicated GS for a child with a suspected genetic condition. We describe ways in which participants articulated who their family included, a crucial construct to understand when describing family-level implications. Our findings can strengthen the design and interpretation of quantitative preference research by allowing a more complete understanding of potential sources of preference heterogeneity and advancing knowledge of the family-level value of GS. In clinical care, our results can guide genetic counselors to better consider relevant aspects of the family context when GS is recommended.

## 2. Materials and Methods

We conducted semi-structured interviews with parents and other primary caregivers (hereafter, “caregivers”). Interview recruitment procedures are explained in detail elsewhere [20]. Briefly, we used purposive sampling to invite caregivers to participate in an interview based on their child’s GS status following clinical evaluation for a suspected genetic condition at a pediatric outpatient clinic. Invited families included those who were either recommended to have GS or had completed GS and received results within the past year. This information was obtained via a review of electronic medical records (EMRs) and clinic administrative records. Interviews were conducted via telephone or videoconference, and all study materials were available in English and Spanish.

To inform the development of the interview guide, we reviewed qualitative studies of parental perspectives on GS that were intended to guide preference research [18,19,31,32,33]. After obtaining informed consent, we asked caregivers about their child and their decision-making regarding GS, including how they thought it might impact their family, factors they considered when deciding whether to have the testing, and the most important aspects (anticipated or realized) of having the results. We asked each participant whom they considered to be part of their child’s family. After the interview, each participant was asked to complete a brief online survey to assess demographic and health-related characteristics. Participants were compensated with a $50 electronic gift card.

Interviews were recorded and professionally transcribed. Spanish-language transcripts were translated to English, and all transcripts were checked for accuracy and de-identified prior to analysis. We analyzed qualitative data in multiple rounds of coding using thematic analysis [34]. We developed a consensus coding scheme based on a sample of six transcripts, which was iteratively refined as three members of the study team applied it to the remaining transcripts. Based on the coded data, we developed higher order themes and domains. We used MAXQDA (VERBI Software 2021, Berlin, Germany) and Microsoft Excel to facilitate coding, analysis, and data management. This research was approved by the Baylor College of Medicine Institutional Review Board (H-48379).

## 3. Results

### 3.1. Participant Characteristics

We contacted 104 eligible caregivers, 59 (57%) of whom responded and agreed to participate. Forty-one participants completed interviews between 1 June and 1 September 2021, which lasted 37 min on average (range: 18–73 min). Thirty-five (85.4%) interviews were conducted in English and six (14.6%) were conducted in Spanish. Participant demographic characteristics are displayed in Table 1 and reflect the genetics clinic population. The mean (SD) age of participants was 36.7 (7.8) years. Most participants were the biological mother of the patient (*n* = 33, 80.5%), were married (*n* = 32, 78.0%), and self-identified as either White or European American (*n* = 17, 41.5%), or Hispanic or Latino (*n* = 15, 36.6%). Approximately half of the participants’ children had private insurance (*n* = 22, 53.7%).

Participants’ children were either recommended for GS (*n* = 20) or had received GS results within the past year (*n* = 21). The clinical indications for GS were broad. Some children presented to the genetics clinic with isolated developmental delays, single system symptoms, suspected syndromic conditions, and family histories of heritable and/or known genetic conditions (Appendix A). The child’s mean (SD) age at the time they were recommended for GS was 4.8 (4.2) years (range: 5 months to 17 years). Only one participant whose child had been recommended to receive exome sequencing had chosen not to complete the testing at the time of the interview, as the child’s condition had been satisfactorily diagnosed by chromosomal microarray (CMA). Among the patients who had received GS results (*n* = 21), eight (38.1%) were diagnostic and thirteen (61.9%) were non-diagnostic.

### 3.2. Thematic Analysis Findings

Table 2 provides an overview of thematic analysis findings. Paragraphs below present results organized by domain and theme.

### 3.3. Family Structure and Dynamics

#### 3.3.1. Family Structure

Caregivers shared how aspects of the structure and dynamics of their family contributed to their experiences with GS for their child (Figure 1). For some, the child recommended for GS was their first, while for others, the child was presumed to be their last, which provided important context for how they understood their child’s undiagnosed condition and the implications of the results for future family decisions. Caregivers who were “pretty sure [this child is] our last,” and were not “worried about having more kids,” did not consider the GS results to impact family planning. Those who definitely or probably wanted more children shared how the GS results shaped their reproductive decisions. One caregiver said the GS results “determined whether we were going to have more children or not, because if it was something hereditary, we were like one kid’s enough. Like we just want to get through her and not have another one. So that was kind of like a reason why we really did it too”. Similarly, another caregiver stated, “Well, it affect[s] us a lot, for example, because she is an only child, this can have an impact as we consider if we can have more children”. Some caregivers were single parents, while others had much larger family structures; one described living in a multi-generational household and having a tight-knit family of more than 15 people. Caregivers also referred to social and geographical distance from their family members, including feelings of isolation with their family support system in another country. A few caregivers were not the biological parents of the child and were either an extended family member (e.g., aunt) or foster or adoptive parent. One such caregiver had provided foster care and adopted multiple children with special needs, many with genetic diagnoses.

#### 3.3.2. Considerations for Other Children and Future Generations

Caregivers considered other children in their decisions about GS. Many were planning to have additional children and were wondering or expressed worry about the chances of having another child who was also affected. GS results provided some families guidance on how to grow their families, either through “reassurance in terms of continuing our family” or consideration for adoption if a future biological child’s condition “was going to be severe the next time”. Beyond the child and future children of the parents, caregivers also thought about their current children, future children of their child, and children in their extended families (e.g., the caregivers’ nieces and nephews). Knowing whether siblings of the child could be carriers or also affected with the same genetic condition was important and was information parents also wanted to know and share with their children. Genetic diagnoses in the child prompted medical and genetic evaluations of their siblings, and caregivers spoke about the results being important for informing care for children in the extended family who had more limited access to GS.

#### 3.3.3. Family Dynamics and Support

The dynamics and support offered by family members colored caregivers’ experiences with GS. Parents considered the relationships between the child and their sibling(s) when considering genetic testing, often seeing testing as a means to help the sibling understand the child and improve their relationship. Testing and receiving a diagnosis was viewed as a way of knowing what to expect in the future for the family, including for siblings who may eventually become caregivers of the child. Some caregivers received emotional support from their family members in the pursuit of GS. For example, one parent spoke about the support she received from her own mother, who encouraged her to “love her [child] and focus on just being with her”. Extended family members, who sometimes stepped up to help when the caregiver was burned out, were often in favor of GS for the child. On the other hand, some families were less supportive, and caregivers faced challenging dynamics and conversations regarding the child’s condition and the prospect of GS. Some participants described arguments with their family members, often grandparents of the child, about whether the child might “grow out of” their condition and thus whether looking for a diagnosis was warranted. Caregivers in this situation sometimes mentioned that having a genetic diagnosis would help their families understand them and the child more, potentially leading to a more supportive family environment. Those who lacked support from relatives often relied on close friends for reassurance that they were doing the right thing and getting their children what they needed.

#### 3.3.4. Family Communication of (Genetic) Health Information

Preferences and patterns of family communication of health information and GS results varied widely across caregivers, reflective of their overall family experience. Some caregivers chose to only communicate about the genetic information and evaluation with their spouse and nuclear families. They expressed that they wished to “tak[e] it all in as a family…before [telling] a whole lot of other people,” viewed genetic information as a “personal thing,” or that they were not concerned about what other family members who were not taking care of the child had to say. Additionally, some caregivers did not want to share information with family members whom they saw as judgmental. Others described themselves as an “open book,” saying they would share results regardless of whether they were positive or negative. One parent explained that while they and their spouse were waiting for the genetic test results, the entire extended family was waiting also. Some caregivers chose to post updates on Facebook as a way of disseminating information to their whole family, and others described informal family gatherings as being a setting for family to ask questions and to share progress about the child and their genetic testing. Within their nuclear families, caregivers wanted to be able to communicate and explain the genetic diagnosis to their other children so that the child’s siblings would understand. Parents also wanted to explain the diagnoses to the child so that the child would better understand themselves and why they may have more challenges than other children. One caregiver saw her child’s diagnosis as private information, and she wanted to allow the child to decide when or if he shared his diagnosis with others.

#### 3.3.5. Family Health History Knowledge

Knowledge of family health history is central to medical genetics evaluations and genetic test result interpretation, including how definitive a diagnosis for a child may be. Some caregivers had extensive and detailed information about extended family members on both maternal and paternal sides while others had very little information. In the case of foster care or adoptive families, caregivers had only secondhand information, if any at all. Having a family history of health conditions, both those similar to and unrelated to the child’s presenting symptoms, motivated caregivers to pursue GS for their child. Caregivers who alluded to conditions such as cancers, heart defects, autism, attention deficit hyperactivity disorder, intellectual disability, and psychiatric diagnoses running in their families often suspected an inherited genetic component and wanted to learn more. The absence of detailed family history information on one or both sides of the family was also a motivator. One caregiver stated that “there’s been [health-related] stories in the family…that past generations have had something similar, but nobody admitted to anything,” expressing frustration that none of her relatives would talk about it. For many, a lack of other members in the family having any health problems or undergoing any genetic testing (i.e., having a “negative” family history) led to surprise when their child began to show symptoms and required care by multiple medical specialists.

### 3.4. Parental Identity, Relationships, and Philosophies

#### 3.4.1. Parents’ Relationship with Each Other and Parental Disagreement

Parental identity, relationships, and philosophies impacted how families experienced GS. In two-parent households who were recommended trio GS (testing of both biological parents alongside the child), caregivers described how their relationship with the other parent impacted their decision making and expectations. Many couples were “synced up” and in agreement on their decision to proceed with GS or their partner simply “went along with it”. Those who had less agreement on the decision to proceed with GS brought up differences in attitudes towards genetics in general and opinions on the types of results that could be learned, including information about themselves. Some female caregivers mentioned wanting as much information as possible while describing their male partners as being “scared” of finding something, seeing it as a stressor or even a conspiracy, or being someone who would not ever want to know information about the future. Caregivers wanted to know if they were carriers for a genetic condition, and some had questions about their family history and wanted genetic health information for themselves to fill in gaps. They saw testing as a “good idea to have the chance to find out” about potential heritable conditions, and they spoke about how the results would impact all of their children and could have meaning for “somebody else somewhere in [the] family”.

#### 3.4.2. Parental Outlook, Perspective, and Understanding of Child

Participants differed in their style of parenting as well as parenting outlook, perspectives, and understanding of their child. One caregiver described herself and her child’s father as “very intuitive and very consistent” parents, while another spoke of herself as “a free-spirited person”. They brought those personality and parenting traits to their decision-making and GS experiences. The sentiment that “knowledge is power” emerged when parents shared that they sought out information through research, and they needed information to make decisions for the child and themselves. They saw value in knowing what to look out for and gaining perspective on the broad spectrum of symptoms and phenotypes for other children with the same condition. Parents also anticipated that having GS results would directly impact their parenting. One parent said, “I think it will help…to guide us as parents in how to foster the life of the child”. Parents wanted to be able to lead the child on the “right path”. Some brought up that they would treat their child “the same way no matter what [GS results] we get,” and “are still taking care of him…still helping him out,” continuing to “live our lives”. Other elements of parental outlook and perspective included being more open minded to difference, adjusting the expectations and wishes they had for their child, trying to focus on the positive, and taking things “one step at a time”. Some caregivers also saw GS as a way to get their child the help they needed rather than focusing on their delays and the possibility they might not achieve the same things as their peers. Knowing the reason behind medical and neurodevelopmental challenges was a catalyst for change in interactions between the child, their parents, their family members, and others as understanding grew. One parent wanted to do GS “so I could understand my boy, that [he] can evolve,” seeing it as an opportunity to “grow as a mother”.

#### 3.4.3. Experience Parenting Other Children

Caregivers brought their experiences, or lack thereof, of parenting other children to both the decision to proceed with GS and how they experienced the results when they received them. For those who already had children, caregivers spoke about comparisons between this child and their sibling(s). Many parents referenced their other children who had typical development and did not require complex medical care, noticing the differences with their child undergoing GS and “[throwing] out the typical timeline of when kids are supposed to do things”. In some cases, caregivers already had a child or children with similar health or developmental concerns, leading to consideration of a possible genetic cause. The experiences of medical evaluations for those children, including genetic testing, impacted parent expectations as well. One foster and adoptive parent of multiple children, most of whom were not biologically related, stated that “every child is different,” as many of her other fostered and adopted children had undergone genetic evaluations. Among caregivers who did not have experience parenting other children, some worried about their ability to parent other children if GS were to show that their first child would require their care far into adulthood.

#### 3.4.4. Caregiver’s Affective State and Physical Health

Undergoing GS for children can be an overwhelming and stressful time for parents. Elements of guilt and blame were raised by many caregivers. Participants shared their affective and emotional responses as well as how GS affected their physical health. In the process of GS and medical evaluations, parents wondered if there was something they could have done differently and if it was their fault. Biological mothers wondered whether there was something that they could have done differently in their pregnancies (e.g., they worried about having not eaten the right things, going into labor early, doing in-vitro fertilization, or not taking the right vitamins). One caregiver recounted that “in that period where I was waiting on the results…at least once a day [I thought] that ‘this is something I did, this is my fault.’” Guilt extended to worrying about passing on a variant to other children as well. Knowing whether the genetic condition came from one side of the family or the other was a contentious issue; a few parents felt blamed by their in-laws. Some described how blame was alleviated when GS results pointed to a sporadic genetic event that was not inherited. Outside of guilt and blame, parents experienced considerable affective responses to the GS testing process, many of which were intertwined with their experience of caring for a child with complex medical needs. They described the process as being “very difficult” and causing stress, anxiety, constant worry, and fear of not being able to handle the future. Some felt they were obsessed with researching, in denial, and were overwhelmed and exhausted. They experienced “scary, worrisome nights” and wondered if they were doing the right thing. For a few parents, the psychological effects reached the level of a clinical diagnosis of depression and anxiety requiring medication, and one parent had considered suicide. Physical symptoms such as migraines and shingles were reported, and caregivers’ well-being was “put…on the back burner”. One parent felt “just not being able to really enjoy” her young child because of her own affective state. Upon learning GS results indicating a genetic condition with which most children are nonverbal, another parent described the heartbreak of possibly “not ever [getting] to hear her physically call me mom or tell me that she loves me”. Multiple caregivers considered their own mortality, acknowledging that they “were not going to be here forever” or were “going to die one day” and would not be around to care for their child in the future. Caregivers also grappled with the possibility that their child’s GS results could have implications for their own health or for their child’s health in the future, with one caregiver sharing “…it [could] cause some stress, it could cause anxiety because you may get some answers that you may not want to know or about your health, basically for both our son and for us as parents”. Affective responses were not all negative, however. For example, some described no longer “having that weight on [their] shoulders,” and one mom said that while they “were upset the first day… we just have to encourage him and try to be positive with him for him not to give up”. 

### 3.5. Social and Cultural Differences

#### 3.5.1. Primary Caregiver Identity and Advocacy Mindset

Family-specific social and cultural differences influenced participants’ decisions to proceed with testing and their experiences throughout the process. Participants often held a primary caregiver identity and an advocacy mindset. Caregivers expressed a responsibility to help their children, feeling it was their “responsibility to make sure that I get him tested and so that we could find out why and what to look for [in the future]” and wanting to “make sure it’s not something we’re missing”. Caregivers expressed desire to “do everything” to help their child and “wanting what is best for [them], like trying to [do] whatever science can help with”. For some, this meant knowing that they had “looked everywhere for answers” through GS. Caregivers saw the potential information or genetic diagnosis to “make sure that me as a parent had exactly what I needed to be a good advocate”. This advocacy perspective included “getting educated on what’s going on” and “being their voice”. One foster parent described her unique role as an advocate in the child’s legal custody and adoption proceedings, including how she kept multiple stakeholders abreast of the status of the child’s GS.

#### 3.5.2. Caregivers’ Professional Experience

Caregivers’ professional experiences, whether related to medical care or not, influenced their decisions about GS. A few caregivers had careers in medicine, research, child development, or biology education. For some, their specialized training was what prompted them to seek out additional specialty evaluations, insist on additional testing, or initiate early intervention services (e.g., “I knew there’s something going on”). A parent who had experience with children with special needs saw the potential to find a genetic explanation as “having a label… as getting her the help that she needs”. A mother who was a pediatrician mentioned that if she and her cardiologist husband did not each have health education, GS would have been “more complex”. For others with unrelated professional backgrounds, GS was framed from their view; one parent shared “I’m a risk professional. I do it for a living, so being proactive… That’s my information. I want to know everything and get ahead of it”. Caregivers also relied on extended family members with medical training as a resource for interpreting and processing aspects of GS by turning to their own parents, siblings, and in-laws who were obstetricians, pediatricians, nurses, social workers, or other related professions for help.

#### 3.5.3. Experience with Genetic Conditions

Participants had varied previous experience with a genetic condition, either in a family member or in themselves. They recounted personal experiences of having siblings and extended family members (e.g., aunts, nephews) with genetic or presumed genetic conditions. Participants recalled observing or learning about their relatives undergoing evaluations by medical specialists and, in some cases, having genetic testing. Some with known genetic or clinical diagnoses in the family, such as Down syndrome or congenital heart defects, expressed that they were “expecting something” genetic to show up for their own child. However, for those whose child was the first to have any medical complications, such as a mother who noted that her son had “been to more doctors in the first two years of his life than I have in my life,” expressed not knowing what to expect. One mother, who had a clinical diagnosis of a genetic condition for which her children was undergoing GS, shared that she was trying to find out more for her children so that they could have more choices in life. She did not want them to be left behind and “in the back” like she was when she was a child and “went all these years…not knowing that I had something other than just [being] hearing impaired”. The GS process was familiar to other interviewees. One foster parent described her experience caring for many children with special needs and genetic conditions, and how GS was a routine experience for the family (i.e., “this is what we do”). Because of this familiarity with the GS process, she had an understanding that GS could be helpful to learn more about what to expect, both for the child in her care and for the child’s biological siblings in others’ care.

#### 3.5.4. Experience with the Health Care System

In a similar way, families had varied levels of experience with the health care system. Some were quite familiar with navigating aspects of specialty care and health insurance, while others described coming from cultures and countries such as Mexico or Cuba where they “didn’t know this existed or any other conditions” and genetic testing was not available or routine. A few female caregivers mentioned that their partners avoid going to the doctor. The female caregivers described their male partners as feeling that “doctors are just trying to make it seem everyone has something” or label them with something, which made their partner skeptical of genetic testing. Some felt that their male partner saw having a genetic diagnosis as having something “wrong” rather than a means of having more information that could help guide care. GS and the medical complexities that prompted the genetic evaluation were sometimes associated with negative interactions with the healthcare system. One caregiver mentioned how painful it was to watch her child in the neonatal intensive care unit. Another participant shared, regarding the geneticists’ analysis, “the way they analyze your daughter, or your child’s features and kind of point out some flaws. It’s definitely tough to hear. That might’ve been tougher than the actual diagnosis”. 

#### 3.5.5. Concerns of Misunderstandings or Discrimination within the Family

Caregivers had concerns about discrimination related to GS and recounted misunderstandings within their family about their child. Parents spoke about individual cultural identities, including one caregiver who was from the Virgin Islands and whose husband was Latino (Salvadorian), who described her family as thinking “it’ll go away” as the child grew older. Another parent viewed Catholicism as having a role in her partner’s family’s tendency to “deny” that the child was affected so that no one could “put…stigma on his son”. The idea of “put[ting] a label” on their child with a genetic diagnosis was raised by many parents. While a few expressed concerns that others might place stigma on their child based on their genetics, a larger number expressed seeing a diagnosis as a way to legitimize the disorder to their families and others and as a catalyst for “educating others that differences [are] okay”. 

#### 3.5.6. Availability of Help with Caregiving

Children who undergo GS typically have complex medical and physical needs requiring extra caregiving, and the availability of help with caregiving impacted how caregivers experienced GS. Some caregivers described having help from the child’s grandparents, extended family (aunts), or a nanny, with tasks such as taking the child to therapies, attending appointments, and taking care of the child while the parent attended to other needs. In one case, a family member became the child’s legally designated caregiver. Parents with less caregiving help were in situations where they were living far from their own family, had experienced multiple nannies quitting, or had partners with demanding jobs that left them with most of the caregiving. Caregivers sometimes saw GS as a way of further identifying what caregiving help would be needed. Participants expressed that they wanted all caregivers, including those in their family, to know the child’s diagnosis so that they could understand them and know how to take care of them.

#### 3.5.7. Life Adjustments to Provide Care

Beyond seeking outside caregiving from family and others, those with children undergoing GS made a variety of life adjustments to care for their child, often driven by getting a diagnosis associated with longer term neurodevelopmental delays that would require life-long support. Interviewed mothers adjusted their lives by taking time away from work (through Family Medical Leave Act provisions), sought other means of income, such as freelance work and remote work, to accommodate more flexible schedules and increased expenses, chose not to re-enter the workforce after maternity leaves, and quit their jobs. Parents changed, or planned to change, their living situations by moving to be closer to other family members for support, moving closer to the hospital if GS revealed a diagnosis requiring frequent medical care, and moving to a house with one story that would be more accessible for the child. Smaller adjustments to family gatherings to protect the child from infection (e.g., hand washing, clothes changing, social distancing) were also mentioned.

### 3.6. Conceptualizations of Family

When asked who they considered to be part of their child’s family (Table 3, Figure 2), most caregivers provided a list of extended family members that included the child’s parents, grandparents, aunts, uncles, cousins, and in some cases, more distantly related relatives with whom they would share information about their child’s genetic testing results with. Alternatively, a few caregivers described only those family members in their nuclear family as being part of the child’s family, some going as far as to exclude those who did not provide care for their child (such as extended family members). A few participants used the household to define the family unit, which included siblings, parent’s siblings, and other children and adults in the extended family who may sometimes reside in the same home. In cases of foster and adoptive families, the conceptualization of the child’s family encompassed not only the child’s biological relatives, such as full- and half- siblings and parents, but the other children living their current home, who may or may not be biologically related to the foster or adoptive parents.

Outside of these narrower and more traditional conceptualizations and definitions of family, a few caregivers described their child’s family in other, unique ways. For some, close friends of the family were considered family and, for others, family extended to individuals in the community such as teachers, developmental therapists, or medical team members such as nurses and doctors who play a large part in the child’s life and care.

## 4. Conclusions

In this interview study with caregivers of children who were recommended for or who had recently completed GS, family-level characteristics influenced caregivers’ decision making about GS for their child and shaped their overall experience of the testing process. Parents and other primary caregivers considered the child’s family members, including themselves, other children, and extended family when weighing the benefits and drawbacks of GS for their child. Participants relied on family support systems, according to their own conceptualization of who that included, before, during, and after GS. In some cases, GS was a familial experience where information about GS was shared openly beyond the nuclear family, while others believed results only concerned the nuclear family.

Considerations of family, specifically whether benefit to family members can ever be used to justify GS for a child, has been the subject of debate. Some have questioned the ethics of returning secondary findings and results of adult-onset conditions to a child’s parents [35], who may themselves be at risk of developing that condition. Since this disclosure could lead to risk-reducing and potentially life-saving preventative treatment in parents and other family members, it has been argued that there is a benefit to the child that their parents be informed of this genetic information as to maintain their health and ability to care for their child [36,37]. Indeed, empirical research supports that parents consider the potential benefit to the family when receiving secondary [38] and/or adult-onset findings [39], though other studies have found that some parents reject the relational aspect of the child’s best interests [40]. Likewise, some have argued that the potential family benefit of analyzing and disclosing adult-onset findings is not significant enough to outweigh the child’s right to an open future and their autonomy to decide whether they would like to know their risk of disease as an adult [35].

Our results highlight important considerations for genetic counselors and other clinical providers when counseling families through the GS process. Ascertaining information about how family is conceptualized and the family’s propensity to share medical information allows providers to more appropriately provide psychosocial counseling and discuss potential health implications for extended family. Encouraging conversation about family-level topics can aid in facilitating decision making for caregivers who may be uncertain or feel stuck when deciding whether to pursue GS for their child. Asking caregivers if or how they anticipate sharing genetic results with other individuals in their family, what their family support system looks like, or how GS results for their child could impact family interrelations, may more comprehensively elicit and address important family-level considerations during pre- and post-test genetic counseling for children undergoing GS.

Moreover, understanding the characteristics of individuals and families that drive decision-making is important to the design and analysis of preference research. Better design of quantitative preference studies, such as discrete choice experiments, can advance understanding of the ways in which preferences may differ across population groups [41], allowing for more precise estimates of uptake and value and for genetic counselors to more effectively address preference-sensitive aspects of GS when it is recommended for pediatric patients. For example, our findings contribute to the literature on ways in which social and cultural characteristics of caregivers and their families can impact how caregivers think and make decisions about GS. Our results can aid in designing more nuanced practice recommendations to help support familial decision-making pediatric genomics. Better assessment of preference heterogeneity can lead to more accurate measurement of the value of GS at the family level.

## Figures and Tables

**Figure 1 children-10-00774-f001:**
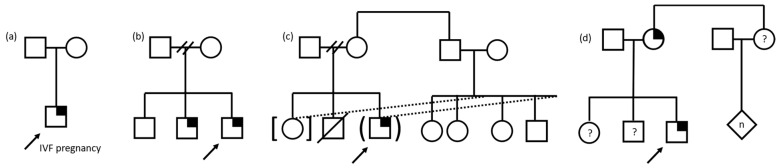
Pedigree examples of family structures influencing family experiences of pediatric genomic sequencing. Child with suspected genetic condition denoted by arrow. Square denotes male, circle denotes female, diamond denotes unknown sex/gender, n denotes multiple and/or unknown number. Symptomatic individual with identified gene variant denoted by quarter shading. (**a**) Only child, conceived through in vitro fertilization (IVF). (**b**) Family structure with separated parents, three sons, the youngest of whom are similarly affected. Two slashes denote separation of relationship. (**c**) Family structure involving foster and adoption of biologically related children resulting in a blended foster/biological family. Dotted lines denote adoption/foster relationship, brackets denote adoption, parentheses denote foster relationship, and slash denotes deceased. Two slashes denote separation of relationship. (**d**) Family structure in which the parent is also affected and the status of the child’s siblings and the parent’s sibling and children is unknown, denoted by question mark.

**Figure 2 children-10-00774-f002:**
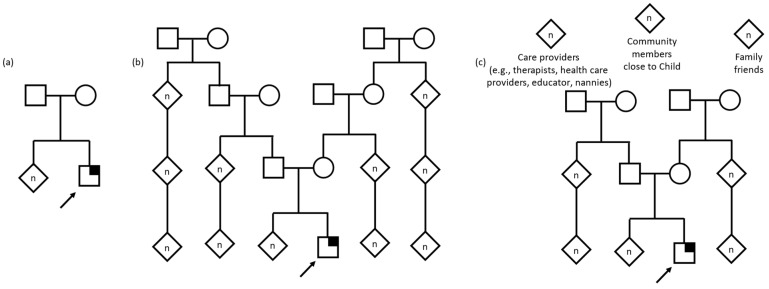
Pedigree examples of family as described by interview participants. Child with suspected genetic condition denoted by arrow. Square denotes male, circle denotes female, diamond denotes unknown sex/gender, n denotes multiple and/or unknown number. Symptomatic individual with identified gene variant denoted by quarter shading. (**a**) Nuclear family defined as a unit limited to parents and children. (**b**) Extended family including multiple generations including grandparents, cousins, great grandparents, second cousins, and others. (**c**) A family defined as anyone who helps take care of or is close to the child, including formal and informal care providers, community members, family friends, and the child’s extended family.

**Table 1 children-10-00774-t001:** Interview participant characteristics (*n* = 41).

	Mean (SD)
Caregiver age	36.7 (7.8)
	*n* (%)
Caregiver’s relationship to child	
Biological mother	33 (80.5%)
Biological father	3 (7.3%)
Legal guardian	3 (7.3%)
Foster Mother	1 (2.4%)
Stepmother	1 (2.4%)
Caregiver self-reported general health	
Fair	5 (12.2%)
Good	13 (31.7%)
Very Good	19 (46.3%)
Excellent	4 (9.8%)
Caregiver gender	
Female	37 (90.2%)
Male	4 (9.8%)
Caregiver’s marital status	
Married	32 (78.0%)
Divorced	3 (7.3%)
Never married	3 (7.3%)
Living with partner	3 (7.3%)
Caregiver’s race and ethnicity	
Asian	3 (7.3%)
Black or African American	4 (9.8%)
White or European American	17 (41.5%)
Hispanic or Latino	15 (36.6%)
Multiracial	2 (4.9%)
Caregiver’s education level	
High school graduate or less	10 (24.4%)
Some college or Associate’s degree	10 (24.4%)
Bachelor’s degree	12 (29.3%)
Graduate or professional degree	9 (22.0%)
Caregiver’s household income	
Less than $40,000	12 (29.3%)
$40,000 to $79,999	14 (34.1%)
$80,000 to $139,999	7 (17.1%)
$140,000 or more	8 (19.5%)
Child’s insurance provider	
Private	22 (53.7%)
Public	19 (46.3%)
Interview language	
English	35 (85.4%)
Spanish	6 (14.6%)
Caregiver-reported severity of their child’s health condition	
Mild	10 (24.4%)
Moderate	20 (48.8%)
Severe	11 (26.8%)
Child’s exome status at interview	
Exome result received	21 (51.2%)
Exome testing submitted	19 (46.3%)
Exome testing recommended but not pursuing	1 (2.4%)

**Table 2 children-10-00774-t002:** Domains, themes, definitions, and example quotes of family experiences of genomic sequencing.

Theme and Description	Example Quote
Domain: Family structure and dynamics	
Family structure: Aspects of family structure, including biological and foster relationships and family planning	“I have three children, three boys. One, the oldest, is age 22. He is in the [Armed Forces], honor roll throughout his whole school career, excelled, was in football and excelled in all these extracurricular activities in school, now lives out of state, is in the [Armed Forces]. So that’s child number one. Child number two is 15 years old, but almost the replica of [child]. So I have two out of three children have special needs sort of parallel. Sort of. Both children, age 15 and [child] are both nonverbal. They are both delayed and have always been in special ed and both have received early intervention and have been in special ed. It’s been big question marks”. (Participant ID 30)
Considerations for other children and future generations: Considerations for the proband’s siblings or for future generations of children (proband’s children or caregivers’ children)	“When I first got the results, it was overwhelming. I had a newborn, when she was only a week or two old whenever I found out that, about his issues. Then, since they are genetic, it’s possible that she had it too. It has made me wary of having any more children, which we always wanted a big family. Yes, it has, but at the same time, it makes me happy that we can treat him and know what to look for in our daughter as well”. (Participant ID 84)
Family dynamics and support: Family interpersonal dynamics, including relationship with the proband, and psychosocial support from family members	“In this country I only live with my husband and my mother-in-law, I don’t have my mother here, she is in Cuba, and one’s mother is always very supportive, especially when you have children. So, I don’t have her here with me. I am literally alone here, and just the family that I created with my husband, my son, and my mother-in-law and her brothers, and so on”. (Participant ID 26)
Family communication of (genetic) health information: Propensity to share genetic or health information with family members	“We don’t really tell family about his diagnosis, so there can’t be discrimination because family lack of knowledge because my extended family would discriminate. So we don’t share with my extended family that he has a diagnosis because they are very discriminatory. As far as like my immediate family, which is the people that would affect as far as if they have something, my husband shared that because he wanted to know if there was someone else who might have that information that would need to know”. (Participant ID 1)
Family health history knowledge: Access to information about, or knowledge of, family health history information	“I think just knowing that we did have a niece with a heart condition that kind of like flagged us thinking, okay, well maybe it is something hereditary and just so and our parents both have heart problems. We wanted to make sure that, that it, like I said, it wasn’t something hereditary. Cause we knew that history, but it wasn’t like her condition is not something that they had. We were wondering if maybe it was influenced by their genes, our genes. So that’s kind of a thought we had”. (Participant ID 114)
Domain: Parental identity, relationships, and philosophies	
Parents’ relationship with each other and parental disagreement: Parents’ relationship with each other or parental disagreement regarding their child’s health condition or genetic testing	“He’s just hesitant when it comes to anything. Everything that has to do with personal information, he’s just very cautious, especially when it comes to medical. His parents have had a lot of medical issues and he grew up dealing with medical system in a different way. With him, it’s just giving assurances of why we’re doing this and what the outcome could be. I think a little bit might be the hesitancy that there could be some bad information. We differ on that. I’d rather have the information. I’d rather know. And I think for him, it’s his son, it’s a little hard to accept that there could be something bad that comes out of it. So there’s a lot of hesitancy there. But he always, I will say, comes around to my side. Talking to medical professionals and hearing their reasoning behind it was good enough. And him knowing I wanted to know made it easy. We didn’t have to fight about it”. (Participant ID 61)
Parental outlook, perspective, and understanding of child: Parent’s outlook and perspective, including understanding of their child and their style of parenting	“That’s actually why I began the whole journey of finding out his diagnosis, because, well, we have family, grandma takes care of him sometimes, dad takes care of him and I just feel like the more we know and the more we can understand [child] and have only perhaps a little more compassion. For me, it wouldn’t change what his diagnosis is for me personally, but sometimes that does help other family members and other people that interact with him to say, ‘Oh, okay. He is pulling hair because this is the tendencies of this diagnosis or he is talking to the floor because this is kind of what they do.’” (Participant ID 30)
Experience parenting other children: Caregiver’s experience with parenting other children	“I can tell you, he’s seven years old and our biggest win for us, because as [child] got older, we threw out the typical timeline of when kids are supposed to do things. I have a three year old who is like night and day from [child] in the sense of he did things that we never got with [child]. Learning how to crawl, learning how to walk. My little monkey climbs up in my cabinet, when you’re not getting the fruit snacks quick enough, he climbs up in there and those are things that [child], forget it. We think about just him and daily activities”. (Participant ID 47)
Caregiver’s affective state and physical health: Affective response and physical impact on caregiver of parenting a child recommended for GS, including parental guilt or blame related to their child’s health condition, broader implications of caring for their child, or genetic testing for their child	“At first I was super excited to receive the call because she had said that there is a diagnosis and that made us happy because we know now what it is that she has. And hopefully, we’ll be able to get her the help and the things that she needs. And then I was a little heartbroken because what [child] has, has only recently been discovered in 2020. There’s not a lot of information on it. There’s a lot of these people with [child]’s syndrome, they don’t speak. And so, just to hear that and think that she, I might not ever get to hear her physically call me mom or tell me that she loves me and stuff like that was, it was really disheartening. And I’m still worried about her to this day on just things in life prom and high school. And who’s going to take care of her after me and her dad are gone and, but I’ve come to accept it. The geneticists helped me find a group that has other parents with children with the same syndrome. There’s not many of us, but we share all of our information and I believe they all did the same testing to receive their diagnosis as well”. (Participant ID 78)
Domain: social and cultural differences	
Primary caregiver identity and advocacy mindset: Perceiving self as the child’s primary caregiver and advocating on their behalf	“Yeah, because once we know exactly what we dealing with, I want to be a part of it. As the caregiver, I want to be able to do what I can to make it a whole lot easier for him, make it more comfortable for him. I would be impacted. I’m already impacted. I’m here for the long haul. I’m not going anywhere”. (Participant ID 68)
Caregivers’ professional experience: Caregivers’ professional training or work experience shaping how they approach caring for a child with a (genetic) health condition	“Even some of my own [students’] parents I used to work with, having a label, being labeled as such, and maybe people predetermining how she is going to be. But I guess I see it so differently, just from my experiences. I don’t see that at all. I see it more as getting her the help that she needs, and making her unique; that’s how I look at it”. (Participant ID 12)
Expereince with genetic conditions: Drawing upon insights from personal or family members’ experience with a complex (genetic) health condition	“My brother has Down syndrome, he has Down Syndrome, and my brother had genetic studies done in the year he was born. We are younger than him, we both [had genetic studies that showed] we did not have Down or any other type of genetic problem. Obviously 45 years ago it wasn’t the quality of genetics it is now, right? However, yes it is related to genetic studies, and also to my parents, for example, it determined a lot how much they could expect from my brother”. (Participant ID 75)
Experience with the health care system: Caregivers’ experience with the health care system (trust, willingness to seek care, affordability, access)	“Well, I’ve always thought, or I’ve always felt like there’s something going on in our family with our genes and stuff. We have a lot of issues, especially on my side. So I didn’t know if some of those issues were the cause of it. So I wanted to know, I wanted to know if any of that was it, because being Hispanic and coming from like my community doesn’t really look into stuff like that. For one, a lot of people can’t afford it or they just, they don’t know about it. They don’t have that knowledge that it’s out there and that it can help you understand what’s going on with you. So, I mean, it was important for me to know, because if I ever did want to have more kids and it was something that was going to be passed on, then I wanted to be able to make that informed decision”. (Participant ID 98)
Concerns of misunderstandings or discrimination within the family: Concerns about discrimination or misunderstandings related to the child’s health condition within the family	“[I]t was very difficult to explain why he could not go outside in 75 degree weather. A lot of people, there was a case [where] my mother-in-law planned a party and she was upset that we had to keep him in the house the whole time. And so with that information, I can say, ‘No, this is a legitimate disorder that he has.’ And we were advocates, we were able to kind of fight that battle because it wasn’t just, ‘Oh, he’s tired. Oh, he’s overheated,’ it’s, ‘No, he really can’t be out there that long.’ So we definitely share with our family. Some of our family is not as receptive to it because they don’t understand it and they’re not personally impacted by it so it is harder sometimes to explain when they don’t really be… They look at him like he’s a perfectly normal kid because everybody wants to see a perfectly normal kid, so we do share it and explain and educate so that they understand why we make some of the decisions, why we can’t attend functions, things like that”. (Participant ID 85)
Availability of help with caregiving: Availability of family members or other caregivers to help with the child’s physical caregiving needs	“I used to have good help for her. I’m not too concerned. It’s mostly having a type of support system set up for her in case anything happens to me that, from what the doctors have told us and from her test results, seems like she’ll be able to function okay. But not having a plan makes me nervous. As far as we have done is just talk to family members to see in case this happens, we want you to take these steps for us in order for her to keep receiving the care that she gets now”. (Participant ID 111)
Life adjustments to provide care: Life adjustments (e.g., quitting job) to provide care for child or siblings	“Oh yeah. I quit my job. I mean I put a three-month notice in to my job saying I need to either go down to part-time, or … Because now our lives are like, I need to be closer, I need to spend more time with him, I need to have time … I can’t just rely on, I can’t take him to some appointments. He needs to go to his speech twice a week, he needs to go to this twice a week, I need to be closer to home to make sure that’s happening. My work definitely has been put on the back burner for me since … Yeah”. (Participant ID 90)

**Table 3 children-10-00774-t003:** Family definitions.

Conceptualization	Example Quotes
Nuclear	“It’s just me, his dad and his siblings. We’re the only ones with him all the time”. (Participant ID 22)
Extended	“My husband and I both come from pretty big families. I always tend to think of our family as like us and then like grandparents, aunts, uncles, cousins, basically. A little bit of that extended family because we’re just so close. I guess he has a lot of Italian roots and then I have some Mexican American roots. So, I feel like some of that is maybe cultural, and that our family is the big family”. (Participant ID 12) “Everyone…My mom, my in-laws, his aunts, uncles, his cousins, my grandmother, his grandmother, everybody. We all are in it with him together. Everybody prays for him. He has a strong village. He has an awesome village”. (Participant ID 43)
Household	“Really I think of what I have in the house. It’s me, my sister, aunt who cares for her, it’s her other sister, [sister’s name], who’s four, and my youngest sister is living with me until she goes to college this summer. That’s it. She has her Mimi… my mom, but my mom did not accept to it, and lets [child] do whatever. She wants to be the fun person to be around. Of course she has her dad, too, but her dad’s not involved. When he calls, I’ll let him see the kids or whatever, but it’s not something… he doesn’t see them every month, or anything like that. It’s just sporadic. When you say family, my household”. (Participant ID 76)
Foster/Biological	“My husband and I and our other kids….[Have] five [children], now. We had six last week, but one of our kids went home. […] Two are biological”. (Participant ID 6) “I feel like we were his intermediate family now. But the first thing that comes to mind will be his biological… His parents and his siblings… And now that [child]’s with us, he has more family members, it’s me and my husband and then we have my 13-year-old daughter, my 10-year-old daughter, my five-year-old daughter, and then [child]’s sister who is four. And then I have a son who’s two, and then [child]’s the last one, who’s one. So yeah- we’re pretty big”. (Participant ID 72)
Friends	“So there’s the four of us, and then there’s my parents. So, my dad and my step-mother, my sister. My brother’s in Taiwan so he doesn’t really have any idea, I don’t think, what’s really going on with him. [dad]’s dad, but I don’t think [dad] told his dad about the testing probably, just so as to kind of not to worry him since he doesn’t really see him on a regular basis. Then we have a family friend. This is actually the mother of his first nanny. She stayed in his life, but she ended up going to school full-time. This woman is really attached to [child] and [child]’s wellbeing, so we do tell her a lot of what’s going on. So I’d say I would consider her also”. (Participant ID 37) “I mean he has tons of people. People who aren’t even classified as family are our family because that’s [child] and it’s not worth it to argue because he’s just like, ‘No, she’s my aunt.’ I’m like, ‘Well technically…’” (Participant ID 47) “I guess, people that we see on a semi daily basis is what I consider family. My best friend who lives down the street, she’s been there from the beginning, so I consider her family. Just someone that’s always been in his life constantly, and has been there while we navigated through all of this, whenever he was first diagnosed with a hearing loss and then the abnormal MRI and everything else that followed”. (Participant ID 104)
Community	“Anyone who is close to him and knows him well. We do have our close friends that we have shared the information with and caretakers, people we’ve grown close to at daycare and all his immediate family is close by, his grandparents, his aunts and uncles. Then, of course he doesn’t have a lot of cousins yet and they’re too young, so they won’t, they’re not, they won’t know, but that’s who we consider to give information to”. (Participant ID 84)

## Data Availability

The data presented in this study are available on request from the corresponding author. The data are not publicly available to protect participant privacy.

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
