# Peer review of "Framing the Family: A Qualitative Exploration of Factors That Shape Family-Level Experience of Pediatric Genomic Sequencing"

_children, 2023, doi:10.3390/children10050774_

Round 1

Reviewer 1 Report

Thank you to the authors for sharing your qualitative study exploring family frameworks and potential influences on pediatric genome sequencing. All comments are from this reviewer's perspective; notably one who appreciates qualitative research with interviews and thematic analysis. Suggestions are largely to improve readability and highlight your findings. While numerous, none should be onerous.

From this reviewer's perspective, the "Framing the Family" manuscript would greatly benefit from far less complex sentence structure and by separating points. Many of the important points raised get buried by long lists within sentences that have an abundance of commas, ands, ors.

Some specific examples include, but are not limited to, those below: 

Authors: typo with SP: and PhD

Line 36-38: Excess of commas, some with questionable location.

Line 46:  “However” does not seem to align with surrounding clauses.

Line 44-51 Excess of “in addition”, additionally”, “Moreover”

All of Intro also has too many “as well as”

Line 53 Typo/repeat “as well as well as”

Line 70 Citation needed (recommend recent scoping)

Lines 70-74; 75-78; 88-92 Too much compounding sentence structures

Line 117  Please specify whether “clinic population” is general pediatrics or pediatric genetics or a composite of pediatric subspecialty clinics (line 90 says pediatric outpatient yet unclear if general or by specialty divisions or programs.)

Line 132 Only mean age is given. Please also provide age range

Table 2 Recommend substituting another word for “mentality” as this term can have a derogatory connotation.

Table 2 Readability is very challenging as this table is currently structured. One suggestion is to reorganize such that the table only has two columns: with “Theme: theme description” together (separated by colon perhaps) in one relatively narrow left hand column and with the quotes encompassing a very wide right hand column. Currently the Domain is obscured, so one suggestion is to capitalize and embolden the Domain at the top of column with relevant themes. Also, currently difficult to read examples in long narrow column and seems disconnected from the Domain and Themes

Table 2 Theme of ”Family Dynamics and Supports” yet Theme Description only notes “psychosocial support from family”? What about other modes of support, such as practical support as well? 

Table 2 Examples: Rather than frequent distractions with “(child’s name),” consider either a fictitious initial or just “(child)”

Table 2 Suggest using parallel statements throughout, such as if theme states “misunderstandings and concerns” the theme description should not state “concerns and misunderstandings

Line 142-144  Needs to be supported by evidence (or a Discussion point) “For some, the c367hild recommended for GS was their first, while for others, the child was presumed to be their last, which provided important context for how they understood their child’s undiagnosed condition and the implications of the results for future family decisions.”

Line 183 “…seeing testing as a means…” Unless everyone stated this, tempering is suggested (i.e., “often” seeing testing as).

Lines 186-195 transitions are recommended from this reviewer’s perspective to reduce  sentences being overly disjointed. Moreover, a few additional clauses are suggested to help bridge disconnects between several sentences.

One example, in midst of points regarding sequencing, this sentence (lines 186-188) appears random – that is, unless “support” refers to ES and then this needs to be made very explicit to the reader--: “Some caregivers received support from their family members, while others faced challenging dynamics. One parent spoke about the support she received from her own mother, who encouraged her to “love her ……”

Line 201-207. Overly compounded sentence. 

Line 240-241  Please edit throughout for extraneous commas: for example: “In two-parent households, who were recommended trio GS that involves testing of both biological parents,….”

Line 270-274 Another example of many that this reviewer feels would benefit from breaking up ideas into smaller segments. Easier to absorb good information if in smaller packets. “Other elements of parental outlook and perspective included being more open minded to difference compared to others, adjusting the expectations and wishes they had for their….. a way to get their child the help they needed rather than focusing on their delays and the possibility they might not achieve the same things as their peers.”

Line 334-336. Recommend unpacking some ideas and not linking unless making the association apparent. For example, combining not wanting to “miss anything“ and do everything I can” with “being their voice….” 

Line 329, 332. Recommend substituting another word for “mentality” as this term can have a derogatory connotation

Line 351-2. It is unclear to this reviewer and seems a leap lacking evidence (citation) to suggest that the comment “to know everything and get ahead of it” is related to being a risk-assessor or non-medical professional. Many parents and parents-to-be have similar comments, including those in medical professions. 

Line 359-362 From this reviewer’s perspective, the frequency “ands” and “ors” dimishes the readability of this sentence.

Line 364-367  Sentence structure beginning with “Contrastingly” seems to this reviewer overly complex.

Line 388- 391 Unclear as written if these two experiences are related: One caregiver mentioned how painful it was to watch her child in the neonatal intensive care unit while another hared that the ….”

Line 407-410 Overly complex sentence and unclear connection obscuring individual points. Please clarify if there is a reason linking “some caregivers described having help… parents, extended family (aunts), or a nanny, with tasks like taking the child to therapies, attending appointments, taking care of the child while the parent attended to other needs,“  with this last clause  “and in one case, being the child’s legally designated caregiver

Table 3 ease of Readability would be improved if left justified rather than centered text

Line 500-503 From this reviewer’s perspective, the next to last statement feels like a reach and would benefit from  tempering or further clarification or some supporting evidence: “For example, our findings that demonstrate how social and cultural characteristics of caregivers and their families can impact how caregivers think and make decisions about GS can be used to design more nuanced practice recommendations to help support decision making in this context." One suggestion is to temper "demonstrate" in light of this being a relatively small sample size and since there exists a body of research and literature of social network theory and caregivers of children with genetic conditions (i.e. Koehly) as well as many patient/family narratives from the rare disease community, The authors may consider noting that your study adds to the literature and further highlights a need to understand these features.

Thank you for the opportunity to review this manuscript and am happy to comment on a revision should the authors and editors wish.

Reviewer 2 Report

I found this paper very interesting and informative, particularly Table 2 .  The data will be a great resource to understanding family dynamics and attitudes towards genetic testing.  Also, the manuscript is very well written 

Recommendations:

Remove "Army" and replace with [employment] to protect the identity of the individual. 

1.     What is the main question addressed by the research?

The authors have carried out interviews with parents/primary caregivers of paediatric patients of a suspected genetic disease to investigate their experience with genomic sequencing (GS). The aim was to understand the definition of a family included extended members that were not related by blood and what their experience/preferences were within the health care system with respect to the genetic condition.

2.     Do you consider the topic original or relevant in the field? Does it 
address a specific gap in the field?

Yes.

Genomic sequencing is readily available and used in clinical decision making.  Thus, the manuscript aim and data analysis was very interesting, particularly the findings. 

The data adds to the field but not a significant gap based on the references cited by the authors.

3.     What does it add to the subject area compared with other published 
material?

Unique, deeply personal, honest, raw comments from family members about their experience. Also, participants come from a range of different income and educational levels.

4.     What specific improvements should the authors consider regarding the 
methodology? What further controls should be considered?

I thought the methodology was described appropriately and could be replicated. The study also received ethical approval from a reputable source.

I originally thought the sample size was low but changed my mind due to the nature of collecting the data (interview time and online survey) and they had two cohorts (i.e., considering GS vs. having, with around 41 applicants).

Perhaps the genetic disease being tested could be declared but ethically, this is not part of the research aim and could introduce bias to which comments have been selected for quotation in the manuscript.

5.     Are the conclusions consistent with the evidence and arguments presented 
and do they address the main question posed?

Yes. The authors have produced clear figures and tables to show the family dynamics and the attitudes of GS.

The data will be helpful for clinicians, genetic counselling and postgraduate students who work with children undergoing GS. Knowing how families and caregivers work/think and also what are the social/cultural issues, will aid designing better practises to support decision-making.

6.     Are the references appropriate?

Yes.

Reviewer 3 Report

Dear Authors,

It was a pleasure to review your article: "Framing the family: A qualitative exploration of factors that shape family-level experience of pediatric genomic sequencing". This is a very well written article that provides relevant information for pre-test and post-test counselling families of children requiring genomic sequencing. 

I found particularly interesting the examples presented for each category. It may b interesting to add an example of incidental findings and how these are perceived by the affected families. 
